# Crystal structure of human U1 snRNP, a small nuclear ribonucleoprotein particle, reveals the mechanism of 5′ splice site recognition

Yasushi Kondo[†‡], Chris Oubridge[†], Anne-Marie M van Roon, Kiyoshi Nagai*

Structural Studies Division, MRC Laboratory of Molecular Biology, Cambridge, United Kingdom

**Abstract** U1 snRNP binds to the 5′ exon-intron junction of pre-mRNA and thus plays a crucial role at an early stage of pre-mRNA splicing. We present two crystal structures of engineered U1 sub-structures, which together reveal at atomic resolution an almost complete network of protein–protein and RNA-protein interactions within U1 snRNP, and show how the 5′ splice site of pre-mRNA is recognised by U1 snRNP. The zinc-finger of U1-C interacts with the duplex between pre-mRNA and the 5′-end of U1 snRNA. The binding of the RNA duplex is stabilized by hydrogen bonds and electrostatic interactions between U1-C and the RNA backbone around the splice junction but U1-C makes no base-specific contacts with pre-mRNA. The structure, together with RNA binding assays, shows that the selection of 5′-splice site nucleotides by U1 snRNP is achieved predominantly through basepairing with U1 snRNA whilst U1-C fine-tunes relative affinities of mismatched 5′-splice sites.

*For correspondence: kn@ mrc-lmb.cam.ac.uk

†These authors contributed equally to this work

Present address: ‡California Institute for Quantitative Biosciences, University of California, Berkeley, Berkeley, United States

Competing interests: The authors declare that no competing interests exist.

## Introduction

Removal of introns from pre-messenger RNA (pre-mRNA) is an essential step in eukaryotic gene expression. This process is catalysed by a large and dynamic RNA-protein assembly called the spliceosome, which consists of five small nuclear ribonucleoprotein particles (U1, U2, U4, U5 and U6 snRNPs) and numerous non-snRNP proteins (*Will and Lührmann, 2011*). U1 snRNP recognizes a short sequence at the 5′-splice site (5′SS) of pre-mRNA through basepairing between the 5′-end of U1 snRNA and the 5′SS sequence (*Lerner et al., 1980*; *Zhuang and Weiner, 1986*; *Siliciano and Guthrie, 1988*; *Séraphin et al., 1988*) and promotes an ordered assembly of the four remaining snRNPs to form the spliceosome, which then undergoes extensive conformational and compositional remodelings to become catalytically active (*Will and Lührmann, 2011*). During activation, the interaction between U1 snRNP and the 5′SS is disrupted by RNA helicase Prp28 (*Staley and Guthrie, 1999*) and then the 5′SS intron sequence base-pairs with part of the ACAGAGA box in U6 snRNA (*Sawa and Abelson, 1992*; *Sawa and Shimura, 1992*; *Wassarman and Steitz, 1992*; *Kandels-Lewis and Séraphin, 1993*; *Lesser and Guthrie, 1993*; *Sontheimer and Steitz, 1993*); whilst the 5′ exon interacts with U5 snRNA loop I for the first *trans*-esterification reaction (*Newman and Norman, 1992*). U1 snRNP is also an important regulator of mRNA 3′ end cleavage and polyadenylation (*Almada et al., 2013*; reviewed in *Spraggon and Cartegni (2013)*).

Human U1 snRNP comprises U1 snRNA, seven Sm proteins (SmB/SmB′, SmD1, SmD2, SmD3, SmE, SmF and SmG) and three U1-specific proteins (U1-70K, U1-A and U1-C) (*Hinterberger et al., 1983*; *Bringmann and Lührmann, 1986*). We reported the structure of the functional core of U1 snRNP (*Pomeranz Krummel et al., 2009*) based on an experimental electron density map at 5.5 Å resolution to which we fitted previously-determined structures of protein components (*Kambach et al., 1999*; *Muto et al., 2004*; *Leung, 2005*). The most striking feature of the structure is the N-terminal region of U1-70k, which extends from its RRM through a long α-helix and wraps around the Sm protein assembly

**eLife digest** Genes are made up of long stretches of DNA. The regions of a gene that code for proteins (known as exons) are interrupted by stretches of non-coding DNA called introns. To produce proteins from a gene, the DNA is 'transcribed' to form pre-mRNA molecules, from which the introns must be removed in a process called splicing. The remaining exons are then joined together to form a mature mRNA molecule that contains the instructions to build a protein. Errors in the splicing process can lead to numerous diseases, such as cancer.

A molecular machine known as a spliceosome is responsible for splicing the pre-mRNA molecules. This consists of five different complexes called small nuclear ribonucleoprotein particles (snRNPs), which are in turn made up from numerous proteins and RNA molecules. The spliceosome assembles anew every time it splices, and an early step in this assembly process involves the interaction of an snRNP called U1 with the start of an intron in the pre-mRNA. This interaction then stimulates the assembly of the rest of the spliceosome. In 2009, researchers reported the structure of the U1 snRNP, but the structure did not contain enough detail to reveal how the snRNP recognizes the start of an intron.

Kondo, Oubridge et al., including some of the researchers involved in the 2009 work, now present the crystal structure of the human version of the U1 snRNP in more detail. High-quality crystal structures of the complete U1 snRNP molecule could not be obtained because the arrangement of the RNA molecules in the snRNP prevented a regular crystal from forming. Kondo, Oubridge et al. instead engineered two subcomponents of U1 snRNP that each crystallized well, and determined their structures. This revealed that the interactions between the various parts of the U1 snRNP form a complex network.

A protein present in the U1 snRNP, known as U1-C, had previously been reported to be able to recognize introns on its own—without requiring the complete U1 snRNP. Kondo, Oubridge et al. reveal that this is not the case and that U1-C does not read the intron RNA sequence directly. Instead, U1 snRNP is able to find the start of the intron because the U1 RNA can stably bind to this site. The U1-C protein can however adjust the strength of this binding to ensure that the spliceosome can operate with a variety of intron start sequences (or signals).

so that its N-terminus makes contact with U1-C protein, thus accounting for the requirement of U1-70k for U1-C binding (*Nelissen et al., 1994*; *Hilleren et al., 1995*). In this crystal the 5'-end of U1 snRNA pairs with its symmetry-related counterpart, mimicking the binding of the 5'SS of pre-mRNA to U1 snRNP. The Zn-finger domain of U1-C is located adjacent to this RNA duplex but the low-resolution map was insufficient for analysis of the RNA-protein contacts in atomic detail and hence it was not clear how U1-C contributes to the recognition of the 5'SS. The structure of U1 snRNP from HeLa cells, treated with chymotrypsin, was subsequently reported (*Weber et al., 2010*). Although this crystal had a DNA oligonucleotide with the 5'SS consensus sequence bound to the 5'-end of U1 snRNA, the N-terminal end of U1-70k together with U1-C protein were lost by protease treatment. Hence neither of these structures revealed molecular details of 5'SS recognition by U1 snRNP.

In order to gain crucial insight into the mechanism of 5'SS recognition we continued our attempts to grow crystals of U1 snRNP diffracting to high resolution but this proved unsuccessful because the inherent mobility of long RNA helices arranged as a 4-way junction prevented the formation of well-diffracting crystals (*Oubridge et al., 2009*; *Weber et al., 2010*). Hence we designed two sub-structures of U1 snRNP, with exclusively human sequences, based on our 5.5 Å resolution structure and determined their crystal structures at high resolution.

Yeast U1 snRNP, when compared to the human particle, contains a larger and more complex snRNA, which is associated with many protein factors (Prp39, Snu71, Prp40, Prp42, Nam8, Snu56, Urn1 and Prp5), which have no counterparts in human U1 snRNP (*Neubauer et al., 1997*). However, despite these differences, the sequence of the 5'-single stranded region of U1 snRNA (nts 1–10) is invariant from yeast to human (http://rfam.sanger.ac.uk/) and the amino acid sequence of the Zn-finger of U1-C (yeast Yhc1) is also highly conserved (*Muto et al., 2004*). Hence the 5'SS of pre-mRNA is expected to make exactly the same contacts with the 5' end of U1 snRNA and U1-C in human and yeast U1 snRNPs. However, some positions of the 5'SS have quite different nucleotide bias in yeast and human genes

(*Burge et al., 1999*). In human the 5′SS sequences processed by major spliceosomes are degenerate but show significant overall complementarity to the sequence of the 5′ end of U1 snRNA (*Lerner et al., 1980*). In contrast the 5′SS intron sequence of yeast pre-mRNA is stringently conserved to be GUAUGU (*Burge et al., 1999*). Upon activation of the yeast spliceosome, the intron sequence, UGU (+4, +5 and +6), pairs with ACA within the ACAGAGA sequence in U6 snRNA and hence these nucleotides are selected to be nearly invariant (*Sawa and Abelson, 1992*; *Kandels-Lewis and Séraphin, 1993*). Cross-linking studies revealed interaction of the same regions of U6 snRNA and the 5′SS sequence in human but the sequence requirement is less obvious (*Sawa and Shimura, 1992*; *Wassarman and Steitz, 1992*). Prp8 is also known to influence the selection of the 5′SS nucleotide (reviewed in *Grainger and Beggs (2005)*; *Galej et al. (2013)*). Furthermore in humans constitutive or alternative splicing factors facilitate the binding of U1 snRNP to weak splice sites. As discussed above, the 5′SS is subjected to multiple selections which differ in yeast and human and give rise to different nucleotide biases at 5′SS. The strength of variant 5′SS sequences is assessed by relative usage of competing 5′SS (*Roca et al., 2005*, *2012*) which is determined not only by the affinity of 5′SS to U1 snRNP but also by multiple factors (*Roca et al., 2013*).

Our two new crystals together reveal the structures of the substantial parts of U1 snRNP at high resolution and provide crucial insights into the mechanism of pre-mRNA recognition by U1 snRNP. In particular, we find that U1-C makes no base-specific contacts with the 5′SS sequence. Also, by measuring the intrinsic affinity of recombinant U1 snRNP for various 5′SS sequences we disentangle the role played by the U1 snRNP from the other complexities of 5′SS recognition, and assess the relative contributions of U1 snRNA and U1-C protein in light of our crystal structure.

## Results and discussion

A minimal U1 snRNP consisting of seven Sm proteins, the N-terminal peptide of U1-70k, U1-C and a truncated U1 snRNA was designed based on the 5.5 Å resolution structure (*Figure 1A*; *Figure 1—figure supplements 1 and 2*) (*Pomeranz Krummel et al., 2009*). A large portion of RNA attached to Helix H through the 4-way junction was replaced by a kissing-loop (*Ennifar et al., 2001*) to facilitate crystal contacts (*Figure 1—figure supplement 2*). However, this eliminated the binding site (stem-loop I) for the U1-70k RRM (RNA-recognition motif) (*Query et al., 1989*) and thus weakened the binding of the U1-70k N-terminal peptide to the snRNP core domain (*Nelissen et al., 1994*; *Hilleren et al., 1995*). In order to stabilize its binding we fused the N-terminal 59 residue peptide of U1-70k to SmD1 via a Gly–Ser linker (70kSmD1F, *Figure 1—figure supplement 1*). The reconstituted complex was stable and yielded crystals diffracting to 3.3 Å (*Table 1*). The second crystal (U1A70kF-RNA) contains residues 60–216 of U1-70k and the entire U1 snRNA stem-loop I. In order to promote interaction of the long α-helix of U1-70k with the RNA stem we fused it to the U1-A RRM and capped the RNA with the apical loop of stem-loop II such that the α-helix is anchored to the RNA stem (*Figure 1—figure supplements 1 and 2*). The crystal structure of this complex was determined at 2.5 Å (*Figure 1B*; *Table 1*). Phases for both crystal structures were determined by molecular replacement.

The structure of the minimal U1 snRNP fits well into the 5.5 Å electron density except for a slight tilt of the duplex between pre-mRNA and the 5′-end of U1 snRNA (*Figure 1C*) (*Pomeranz Krummel et al., 2009*). In the previous structure the fortuitous interaction between the 5′-end of U1 snRNA from two symmetry-related complexes could have distorted the orientation of this RNA duplex. Hence we believe that the minimal U1 snRNP, with the consensus 5′SS oligonucleotide bound to the 5′-end of U1 snRNA, represents the 5′SS-U1 snRNP interaction in the whole U1 snRNP. The overall structure of the core domain is very similar to that of the U4 snRNP core domain, which consists of seven Sm proteins and U4 snRNA (*Leung et al., 2011*) and the Sm folds of the seven Sm protein assemblies of the two structures (PDB code: 4PJO and 4WZJ) superimpose with rmsd of 0.55 Å. However, there are some noteworthy differences (*Figure 2* and *Figure 2—figure supplement 1*). In the minimal U1 snRNP, helix H0 of SmD2 points into the minor groove of helix H (*Figure 1A*), buttressing it, a feature already evident at 5.5 Å (*Pomeranz Krummel et al., 2009*). The nonamer Sm site sequence (AAUUUGUGG in U1 snRNA and AAUUUUUGA in U4 snRNA) has been reported as a minimal RNA oligonucleotide to promote Sm core domain assembly (*Raker et al., 1999*). In U1 snRNP each base of the first seven nucleotides of the nonamer, AAUUUGU (A125 to U131), interacts one-to-one with SmF-SmE-SmG-SmD3-SmB-SmD1-SmD2 in the pockets formed by four key residues at equivalent positions in the L3 and L5 loops of the Sm fold except for the interaction of G130 with SmD1 (*Figure 2*; *Figure 2—figure supplement 1*). In the protease-treated U1 snRNP structure (*Weber et al., 2010*), G132 was placed in

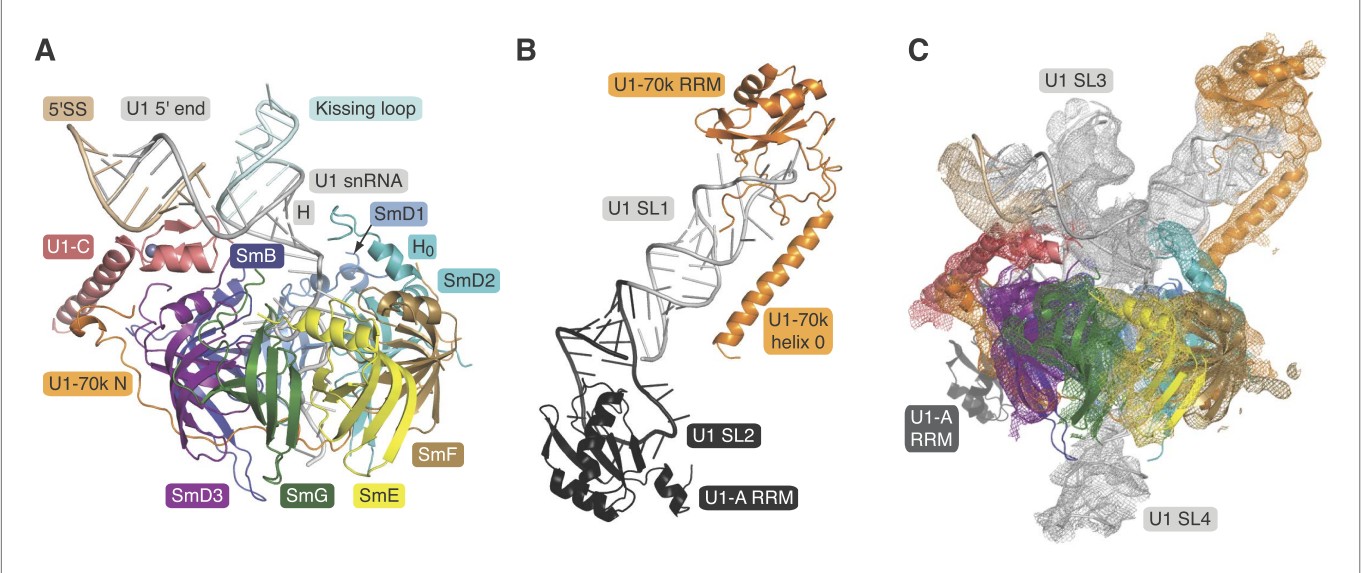

**Figure 1**. Crystal structures of the two sub-structures of U1 snRNP. (**A**) Crystal structure of the minimal U1 snRNP with the 5′-splice site RNA determined at 3.3 Å resolution. Label H indicates U1 snRNA helix H; H0 indicates the first alpha helix of SmD2 protein. (**B**) Crystal structure (U1A70kF-RNA) of the remainder of U1-70k (residue 60–216) bound to stem-loop I of U1 snRNA determined at 2.5 Å resolution. RRM1 of U1A is fused to the residues 60–216 of U1-70k via Gly–Ser linker. Stem-loops I and II of U1 snRNA form a dumb-bell structure. (**C**) Crystal structures of the two sub-structures placed into the experimental electron density map at 5.5 Å of U1 snRNP (*Pomeranz Krummel et al., 2009*).

The following figure supplements are available for figure 1:

**Figure supplement 1**. Protein constructs used in this study.

**Figure supplement 2**. The U1 snRNA constructs used for this work.

the nucleotide-binding pocket in SmF but our structure unambiguously shows that A125 occupies this pocket. The U4 core snRNP structure has now been refined to an $R_{free}$ of 22.4% (Li, Leung, YK, CO and KN, manuscript in preparation; PDB code: 4WZJ) and the new model shows that whereas previously the G equivalent to G132 was also incorrectly placed in SmF (*Leung et al., 2011*) the Sm site nucleotides bind in a similar manner in both minimal U1 and U4 core domain, except for SmD1. The fourth U of the U4 Sm site sequence is replaced by G in U1 snRNA in the majority of species including humans (http://rfam.sanger.ac.uk/) (*Burge et al., 2012*). A large guanine base cannot be accommodated in SmD1 and hence it lies above His37 of SmB outside the central hole (*Figure 2*; *Figure 2—figure supplement 1F*). In the Sm site sequence the phosphate groups of A126, U130 and U131 come close and are stabilized by a hydrated $Mg^{++}$ ion (*Figure 2*). The last two nucleotides of the nonamer, G132 and G133, fit into the binding pocket formed by SmD1-SmD2 and SmF-SmE, respectively (*Figure 2—figure supplement 2*).

In the original 5.5 Å resolution map we roughly modelled the first 60 residues of U1-70k using seleno-methionine derivatives of mutants of U1-70K (L9M, I19M, E31M, I41M, E49M, E61M and I75M). In the 3.3 Å structure almost the entire length of the U1-70k N-terminal peptide is well ordered and its interaction with the core domain is revealed in detail (*Figure 3*). Residues 49–58, containing a type II proline helix (Pro55, Pro56 and Pro57), interact exclusively with SmD2 and its binding to SmD2 is stabilized by salt-bridges and hydrogen bonds (*Figure 3A*). The U1-70k residues 39–49 are wedged in the crevice between SmD2 and SmF (*Figure 3B*). Then the U1-70k peptide crosses the central hole where it interacts with the backbone and bases of stem IV (A135 to G138) of U1 snRNA (*Figure 3C*). The 3′ stem of U4 snRNA (stem II) blocks the path of the U1-70k peptide in the U4 core domain and hence the binding of this peptide is snRNA dependent (*Leung et al., 2011*). The U1-70k residues 10–31 make extensive contacts as an extended peptide with SmD3 and the β4 strand of SmB (*Figure 3D*; *Figure 4A*). The residues 6–12 of U1-70k form a $3_{10}$ helix and together with its N-terminal region contact the long helix B of U1-C to stabilize U1-C binding (*Figure 4A*). Apart from the interaction with

**Table 1.** Crystallographic data collection and refinement statistics

| | Minimal U1 | U1A70kF·RNA |
|---|---|---|
| Data collection | | |
| Space group | $P2_12_12_1$ | C2 |
| Cell dimensions | | |
| a, b, c (Å) | 120.4, 172.6, 256.3 | 80.2, 66.6, 93.7 |
| α, β, γ (°) | 90.0, 90.0, 90.0 | 90.0, 111.0, 90.0 |
| Wavelength (Å) | 0.9795 | 1.0332 |
| Resolution (outer shell) (Å) | 69.67 − 3.30 (3.36 − 3.30) | 44.00 − 2.50 (2.64 − 2.50) |
| Unique observations | 80,571 (4538) | 15,967 (2301) |
| Redundancy | 4.6 (4.7) | 3.7 (3.7) |
| Completeness | 99.5 (99.3) | 99.1 (99.4) |
| $R_{merge}$* | 0.153 (0.846) | 0.076 (0.350) |
| $R_{p.i.m.}$† | 0.100 (0.534) | 0.047 (0.210) |
| Mn([I]/sd[I]) | 172.1 (2.7) | 11.0 (3.2) |
| Refinement | | |
| Resolution (Å) | 3.30 | 2.50 |
| Number of reflections | 74,960 | 15,164 |
| $R_{work}$/$R_{free}$ | 0.209/0.255 | 0.203/0.258 |
| Number of atoms | 26,945 | 3248 |
| Mean B-factor | 99.2 | 49.5 |
| R.m.s.d. bond length (Å) | 0.010 | 0.012 |
| R.m.s.d. bond angles (°) | 1.53 | 1.67 |
| Ramachandran statistics for protein residues ‡ | | |
| In preferred regions | 2534 (96.31%) | 228 (96.61%) |
| In allowed regions | 96 (3.65%) | 8 (3.39%) |
| Outliers | 1 (0.04%) | 0 |

*Merging R factor.

$$R_{merge} = \sum_{hkl} \sum_{i} \left| I_i(hkl) - \overline{I(hkl)} \right| / \sum_{hkl} \sum_{i} I_i(hkl)$$

†Precision-indicating merging R factor.

$$R_{p.i.m} = \sum_{hkl} \left[ 1/(N-1) \right]^{1/2} \sum_{i} \left| I_i(hkl) - \overline{I(hkl)} \right| / \sum_{hkl} \sum_{i} I_i(hkl)$$

‡Calculated in Coot (**Emsley et al., 2010**).

U1-70k, U1-C interacts exclusively with SmD3. Helix A and β1 of the Zn-finger and part of long Helix B of U1-C sit on the concave surface of SmD3 made by the N-terminal helix 1, β2 strand and loop 2 and a short α-helix made by the C-terminal tail. The interface between U1-C and SmD3 is not tightly packed and is stabilized by salt-bridges and hydrogen bonds (**Figure 4A**). The R21Q mutation in U1-C weakens its binding to U1 snRNP by about 10-fold (**Muto et al., 2004**).

Natural U1 snRNA has a tri-methyl-guanosine cap attached to the 5'-end but our U1 snRNA has a 5'-triphosphate resulting from in vitro transcription instead (**Figure 1—figure supplement 2A,B**). It also has uridines at positions 5 and 6 instead of the pseudo-uridines found in natural U1 snRNA (**Reddy et al., 1981**). Pseudo-uridines at these positions are conserved from yeast to human U1 snRNA (**Massenet et al., 1999**). Only poorly diffracting crystals were obtained with pseudo-uridines at these positions. In the minimal U1 snRNP crystal, nucleotides 3 to 10 of U1 snRNA form Watson-Crick base-pairs with the AG/GUAAGU sequence of the pre-mRNA strand (**Figure 4B–C**) where / marks the splice

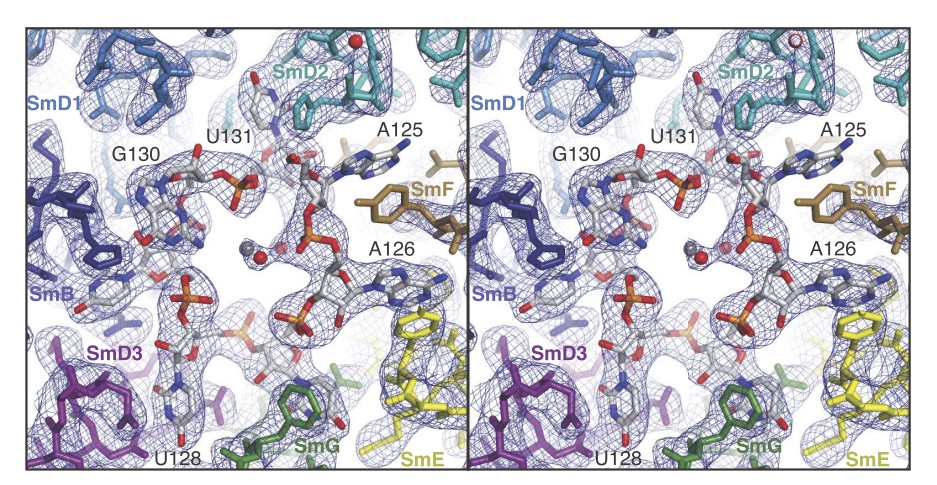

**Figure 2**. Stereoview showing binding of U1 snRNA at the central hole of the Sm protein assembly. The refined model is overlaid onto the 3.3 Å electron density map (2Fo − Fc) contoured at 1.5σ. Electron density for A125 and the phenol ring of SmF Y39, which stacks on it, is weak. Hydrated magnesium ion is found in the central hole (grey, Mg++; red, water). SmD3, purple; SmB, dark blue; SmD1, blue; SmD2, cyan; SmF, brown; SmE, yellow; SmG, green.

The following figure supplements are available for figure 2:

**Figure supplement 1**. Interaction between the Sm site nucleotides and Sm proteins in the central hole of the Sm protein assembly.

**Figure supplement 2**. The nucleotides G132, G133 and U134 in the central hole of the Sm protein assembly.

junction. The minor groove of the RNA duplex between the 5′SS and the 5′-end of U1 snRNA interacts with U1-C on the surface made by Helix A and the β1 strand. Several amino-acid side chains and main chain atoms of U1-C form hydrogen bonds with the 2′OH groups and phosphate oxygen atoms of both strands of RNA near the splice junction (*Figure 4B–C*). However, U1-C makes no contacts with the RNA bases (*Figure 4B*). At an early stage of spliceosomal assembly, U1 snRNP binds to the 5′SS and U2 auxiliary factor (U2AF) and splicing factor 1 (SF1) bind to the 3′ splice site and branch point resulting in the formation of E complex. A double mutant (R28G, K29S) of U1-C fails to enhance the formation of E complex (*Will et al., 1996*). The side-chains of Arg28 and Lys29 are located close to the phosphate backbone of U1 snRNA or 5′SS RNA but their side chains have no density in our map. The 5′-triphosphate of U1 snRNA is close to Glu32 and Asp36 of U1-C and their electrostatic repulsion may prevent stable interaction of the tri-methylguanosine cap with U1-C, thus allowing the pre-mRNA strand to gain access to bases of the 5′-end of U1 snRNA to pair.

## Interaction between U1-70k and stem-loop I

The first 215 residues of U1-70k are conserved well from yeast to human (48% sequence similarity), suggesting that this region has an evolutionarily conserved essential function, whereas the C-terminal region, predicted to be poorly structured, has diverged considerably. Some alternative splicing factors are known to bind to this region (*Labourier et al., 2001*; *Ignjatovic et al., 2005*; *Cho et al., 2011*). The U1A70kF-RNA crystal structure (*Figure 1B*) reveals the interaction between stem-loop I and U1-70k in detail, illustrating a new mode of RRM-RNA interaction. The canonical RRM domain is known to bind with RNA through RNP1 and RNP2 motifs as first observed in the U1A- stem loop II complex structure (*Oubridge et al., 1994*). The RRMs of U1-A and U1-70k have very similar structures (the β-strands and α-helices of the two RRMs superimpose with an rmsd of 0.70 Å) (*Figure 5A–B*) whereas the RNA loops bound to these RRMs have strikingly different structures (*Figure 5C–D*). The U1-A bound RNA loop has an open structure with ten nucleotide bases splayed out (*Figure 5B,D*) (*Oubridge et al., 1994*). Bases of the first seven loop nucleotides show stacking interactions, either with adjacent bases or with protein side chains, while the last three nucleotides are poorly ordered.

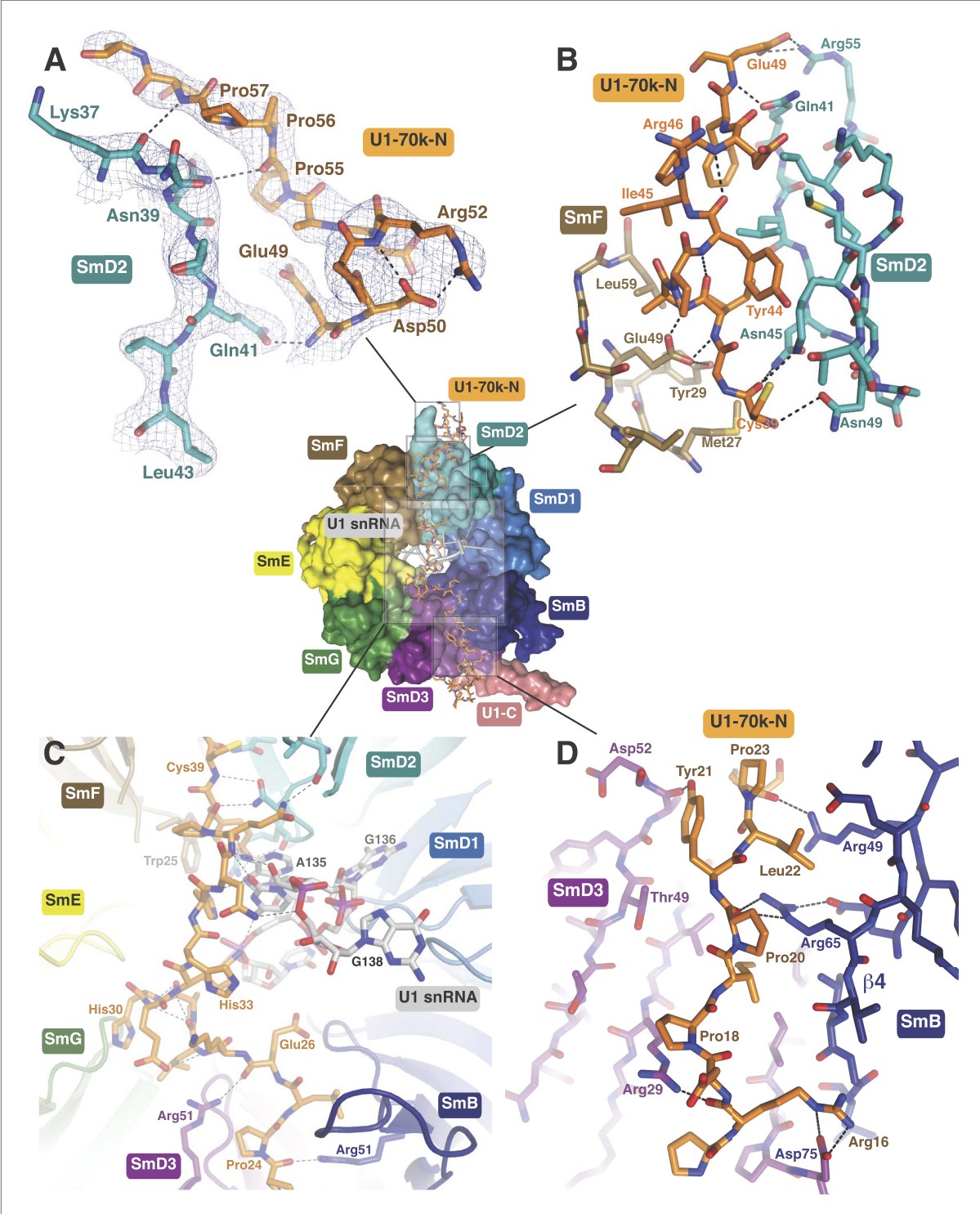

**Figure 3**. The path of the N-terminal peptide of U1-70k. The N-terminal 60 residues of U1-70k run along the interface between SmD2 and SmF, cross the central hole and are wedged between SmB and SmD3 (Inset shows the overview). (**A**) Residues 50–58 interact with loop 1 and β1 of SmD2. Three consecutive Proline residues (Pro54–Pro56) form type II proline helix. (**B**) Residues 39–49 of U1-70k are wedged between SmF and SmD2. (**C**) The U1-70k peptide crosses the central hole where it interacts with nucleotides preceding stem-loop IV. In U4 snRNP the 3' helix is partially buried in the central hole and hence U4 snRNA and U1-70k peptide are mutually exclusive (**Leung et al., 2011**). (**D**) Residues 16–23 of U1-70k are wedged between SmD3 and SmB.

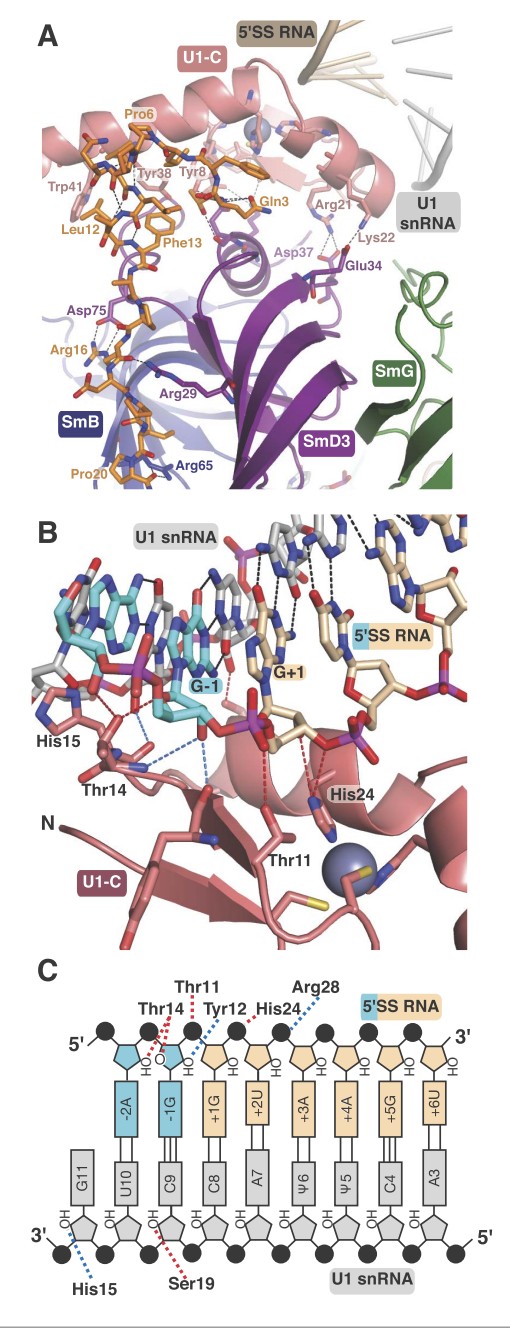

**Figure 4**. U1-C contacts the duplex between the 5'-end of U1 snRNA and the 5'-splice site. (**A**) U1-C sits on SmD3 and its binding is stabilised by the N-terminus of U1-70k. (**B**) U1-C forms hydrogen bonds with the sugar-phosphate backbone atoms but makes no contact with RNA bases. On the 5'SS strand, nucleotides are colored teal for exonic and fawn for intronic sequence. (**C**) Schematic representation of the 5'-splice site recognition. Red dotted lines, hydrogen bonds made by amino acid side chains of U1-C; blue dotted lines, hydrogen bonds made by main chain atoms of U1-C. The 5'SS nucleotides are color-coded as in panel **B**.

In contrast, the U1-70k bound RNA loop 1 with 11 loop nucleotides (*Figure 1—figure supplement 1C*) is stabilized by base stacking interactions and basepairing of nucleotides within the loop and hence it is effectively a five-nucleotide loop (*Figure 5A–C*). In U1-A the polypeptide loop between β2 and β3 (loop 3) protrudes through the RNA loop, stabilising it in an open conformation (*Figure 5B*) whereas loop 3 of U1-70k forms a β-turn and embraces C33 and G34 (*Figure 5A*). The bases of C33 and G34 stack with each other and are sandwiched between the side chains of Arg191 and Lys138, which form salt-bridges with the phosphate groups of C33 and G34 (*Figure 6A*). The second residue of RNP2 motif (Phe106) and the fifth residue of RNP1 motif (Phe148) show stacking interactions with the bases of C31 and A32 (*Figure 6B*) as commonly observed in the RRM-RNA complexes (*Oubridge et al., 1994*). The RNA loop is closed by a trans WC/Hoogsteen base pair formed between A29 and A36 (*Figure 6C*). The base of G28, instead of forming a base pair with G37, flips out from the RNA helix and is sandwiched between the side chains of Arg172 and Tyr112, while the guanidinium group of Arg200 fills the gap (*Figure 6D*). The stacking interaction between G28 and Tyr112 accounts for the UV-crosslinking of these residues in U1 snRNP (*Urlaub et al., 2000*). On the opposite strand the bases A35, A36, G37 and G38 continuously stack (*Figure 6C–D*). U30 is packed against the side chain of Leu175 and forms a hydrogen bond with the side chain of Asp177 and the exocyclic amino group of the adjacent C31 (*Figure 6E*). The stacking interaction between U30 and Leu175 is consistent with the UV-crosslinking of these residues (*Urlaub et al., 2000*).

The most unusual feature of the U1-70k complex is that the regions flanking the RRM fold, which have no apparent secondary structural elements, make extensive interactions with the RNA loop bound on the surface of the β-sheet of the RRM, almost completely burying the RNA loop (*Figure 5E–F*). The C-terminal region following the RRM folds onto the RNA and runs along the shallow minor groove by forming an extensive network of hydrogen bonds. Similarly, the region between helix 0 and the RRM folds onto the RNA.

## Mechanism of 5'SS recognition

The crystal structure of the minimal U1 snRNP has revealed in detail molecular contacts between U1 snRNP and a 5'SS RNA with the consensus sequence. The role of U1-C in stabilising the 5'SS binding was first shown by *Heinrichs et al. (1990)* using 172 nucleotide pre-mRNA. We measured

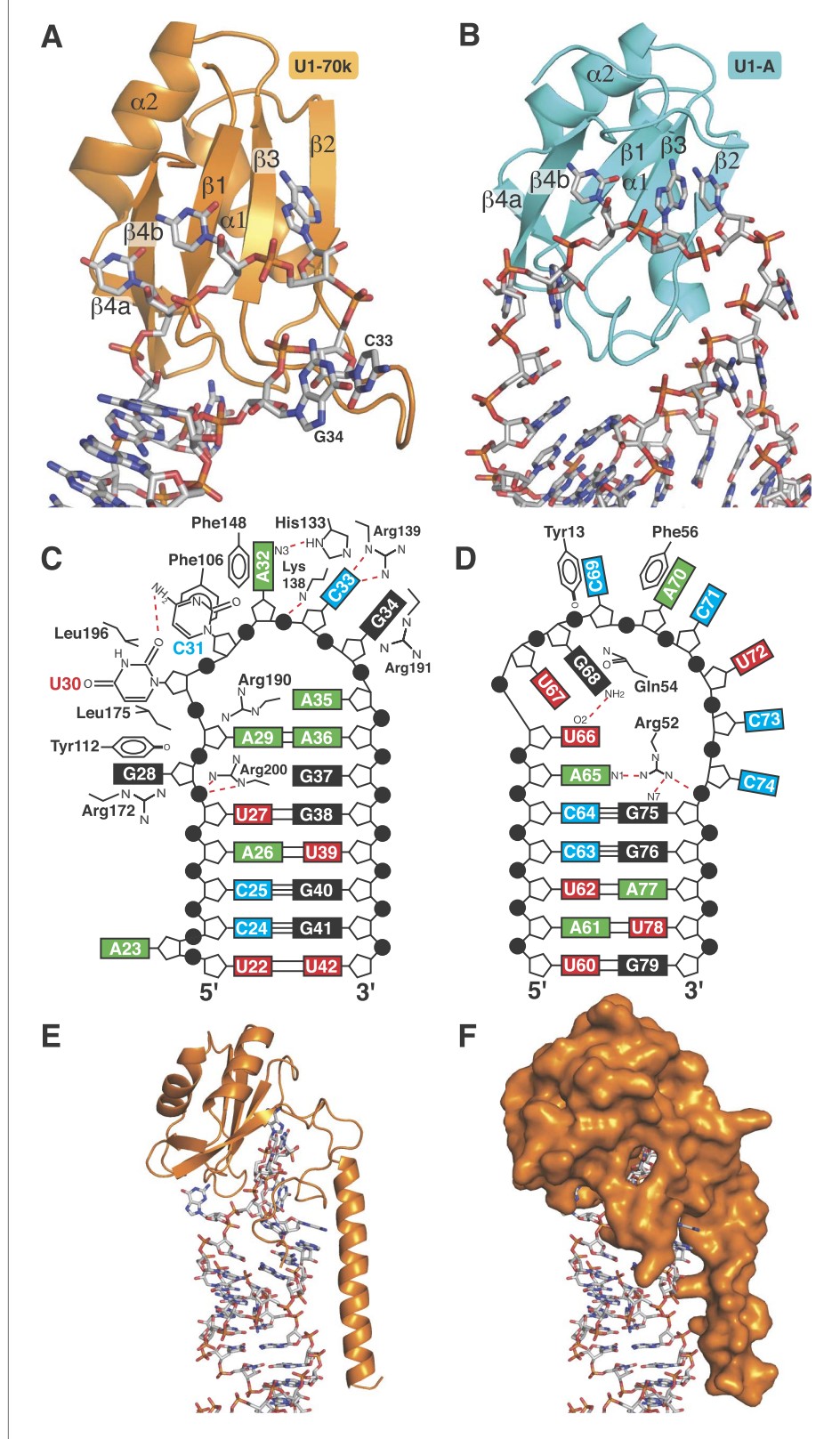

**Figure 5**. RRMs of U1-A and U1-70k show distinct recognition modes of stem-loop I and II. (**A**) Interaction of stem-loop I with U1-70k RRM. (**B**) Interaction of stem-loop II with U1-A RRM. (**C**) Schematic representation of RNA-protein contacts between U1-70k RRM and stem-loop I of U1 snRNA. (**D**) Schematic representation of detailed *Figure 5. Continued on next page*

*Figure 5. Continued*

RNA-protein contacts between U1-A RRM and stem-loop II. (**E**) Regions of U1-70k flanking the RRM folds onto RNA loop and make extensive contacts with RNA. (**F**) Apical loop I is completely covered by U1-70k.

binding of U1 snRNP to a [$^{32}$P]-labelled 5′SS oligonucleotide by filter-binding assay (*Figure 7A*). The affinity of U1 snRNP without U1-C (U1 snRNP[ΔU1-C]) for the wild type 5′SS oligonucleotide increases by between three and fourfold on addition of U1-C (*Figure 7A*). In order to assess the contribution of the molecular contacts revealed by the crystal structure in 5′SS sequence selection we next assayed binding of variant 5′SS oligonucleotides to U1 snRNP containing uncapped, but otherwise fully authentic, U1 snRNA.

Within the 5′SS of human genes processed by the major spliceosome, the most frequently observed nucleotides at each position of pre-mRNA from −3 to +6 form a Watson-Crick basepair with a nucleotide of the 5′-end of U1 snRNA (*Figure 4*; *Figure 7B*). The observed frequency tends to be higher in the middle and taper off towards both ends because mismatches affect the stability of the duplex less when they are further away from the middle. The first two intron nucleotides, which pair with C8 and A7 of U1 snRNA, are nearly invariantly GU (*Burge et al., 1999*; *Sheth et al., 2006*) (*Figure 4C*; *Figure 7C*). The nucleotides at +3 and +4 pair with Ψ6 and Ψ5 of U1 snRNA, respectively, and A/G are found at +3 but A is found predominantly at +4. G is preferred at positions −1 and +5 (*Figure 7B*), which pair with C9 and C4 of U1 snRNA, whilst U is preferred at +6. As the filter-binding assay described above requires large amounts of U1 snRNP, we studied binding of different 5′SS sequences to U1 snRNP by competition assay using a short, fluorescently-labelled oligonucleotide as a reference (*Table 2*). We found that a labelled consensus sequence oligonucleotide bound too tightly to be competed off by the weaker competitor oligonucleotides at concentrations that could be feasibly achieved in the experiment. Hence we used a mismatched labelled oligonucleotide (5SS-F). This does not influence the experiment's capacity to show whether one oligonucleotide competes better or worse than another. Single nucleotide substitutions at highly conserved positions substantially reduce the affinity of the 5′SS oligonucleotide for U1 snRNP (*Figure 7C*). For example, substitution of G at −1, +1 and +5 with C drastically reduces the binding of the competitor oligonucleotide (*Figure 7C*) as C does not form a stable basepair with C (*Leontis et al., 2002*). Furthermore, substitution of +2U with an A severely reduces the binding affinity, presumably because an A–A mismatch distorts and destabilizes the duplex itself (*Leontis et al., 2002*) (*Figure 7C*). The affinity of +2C 5′SS RNA is higher than that of +2A (*Figure 7C*) presumably because cytosine could form a wobble basepair with a protonated A (*Gao and Patel, 1987*; *Jang et al., 1998*; *Wild et al., 2001*). This is consistent with the fact that C is found at +2 position, although very infrequently. G at +3 and +4 could form a wobble basepair with Ψ6 and Ψ5, respectively, and substitution of A with G at +3 or +4 position has only a moderate effect on the affinity (*Figure 7D*). The 5′SS with a G − Ψ wobble basepair at +4 position (+4G) has higher affinity than the one at +3 position (+3G) (*Figure 7C*) whilst +3G occurs more frequently in human genes (*Figure 7B*) than +4G. *Roca et al. (2012)* reported the effect of three single base substitutions in the 5′SS on the melting temperature of the duplex between 5′SS oligonucleotides and an oligonucleotide representing the 5′-end of U1 snRNA. +1A and +2C both had severe effects on melting temperature, in agreement with our results. Most other mutations they studied introduced a bulged nucleotide. Here we did not include such 5′SS oligonucleotides in our analysis, as it is hard to predict the effect of the bulged nucleotide on the molecular contacts observed in our crystal structure. We next studied binding of the same set of the 5′SS oligonucleotides to U1 snRNP in the absence of U1-C (U1 snRNP[ΔU1-C]) to see how U1-C contributes to the selection of 5′SS oligonucleotides (*Figure 7E,F*). To a first approximation these 5′SS mutants show very similar affinities relative to the wild type both in the absence and presence of U1-C, enforcing the notion that 5′SS are selected primarily by U1 snRNA. In the absence of U1-C, the mutant 5′SS oligonucleotides with +3G or +4G compete well with the wild type as in the presence of U1-C but other oligonucleotides with +3U, +4U, +1C, −1C or +5C mutations compete better in the presence of U1-C than in its absence. This shows that U1-C fine-tunes the affinity of these mismatched oligonucleotides, in most sequences studied here, to provide extra stabilisation relative to the wild type. For example, in the presence of U1-C the oligonucleotides with +3U or +4U, which introduce U–U mismatches, compete better with the wild type than they do in the absence of U1-C. Different types of U–U basepair have been reported (*Lietzke et al., 1996*; *Leontis et al., 2002*; *Kiliszek et al., 2009*; *Sheng et al., 2013*). Our

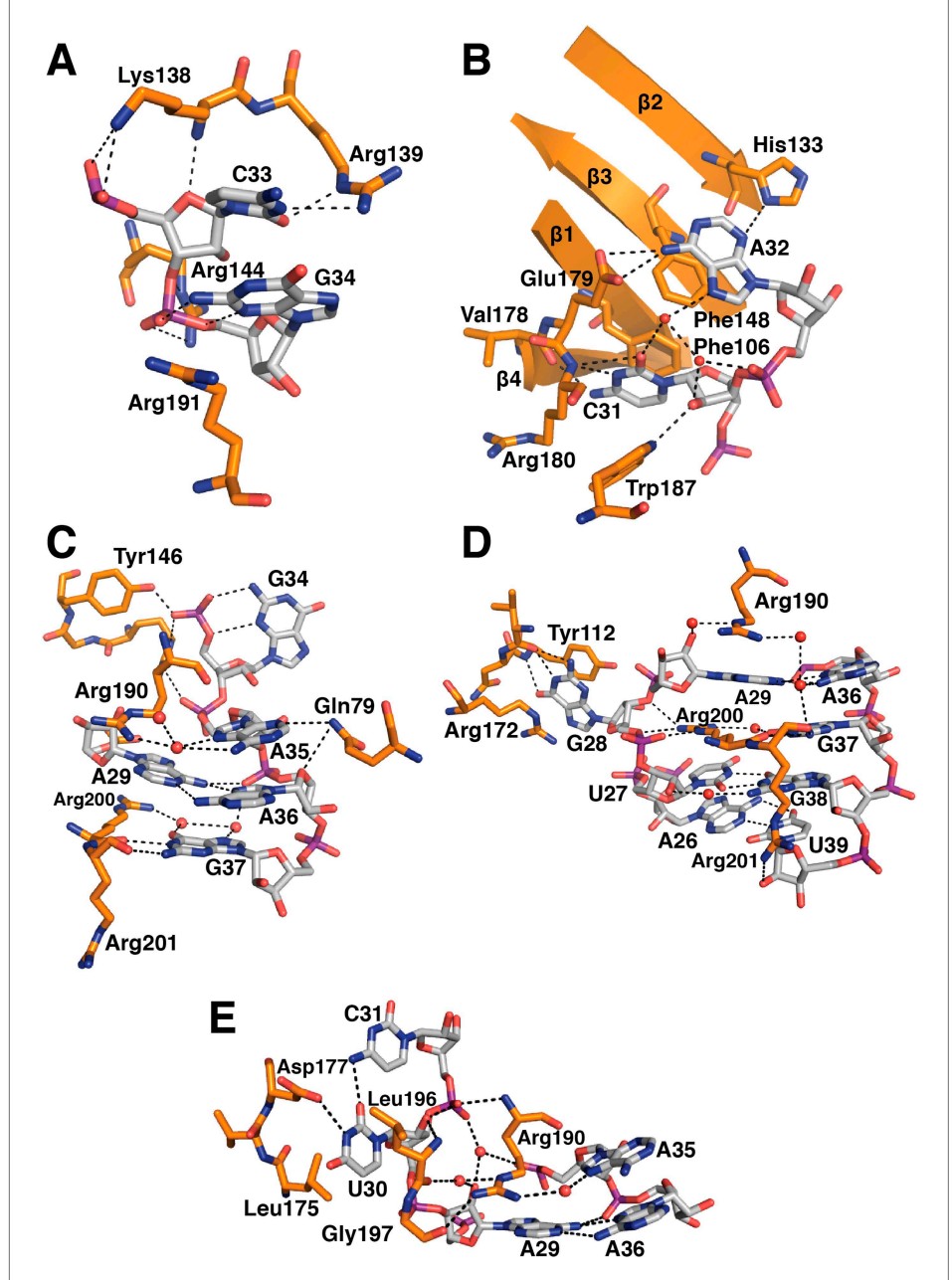

**Figure 6**. Detailed RNA-protein contacts between U1-70k RRM and stem-loop I of U1 snRNA. (**A**) C33 and G34 embraced by U1-70k loop 3. (**B**) C31 and A32 stack onto Phe106 and Phe148 residues of the beta sheet. (**C**) The last three loop nucleotides stack continuously on G38 of the loop-closing base pair. (**D**) The base of G28 is flipped out from the RNA helix and its place is taken by Arg200, which, along with A29 and Arg190, continues the helical stacking of the stem. (**E**) U30 is packed against the hydrophobic side chains of Leu175 and Leu196. In all cases nitrogen atoms are shown in blue, oxygen in red and phosphorus in magenta. Hydrogen bonds are represented as dashed lines. Carbon atoms are coloured grey in RNA, orange in U1-70k.

crystal structure showed that U1-C forms hydrogen bonds exclusively with sugar-phosphate backbone between −2 to +3 positions of pre-mRNA and at C9 and G11 of U1 snRNA (*Figure 4B–C*). When extra hydrogen bonding groups are provided by U1-C, the U–U pair could easily switch from one type to another. It is conceivable that these amino acid residues of U1-C could provide extra stabilization for these non-canonical basepairs, possibly by altering helical geometry and/or minor groove width.

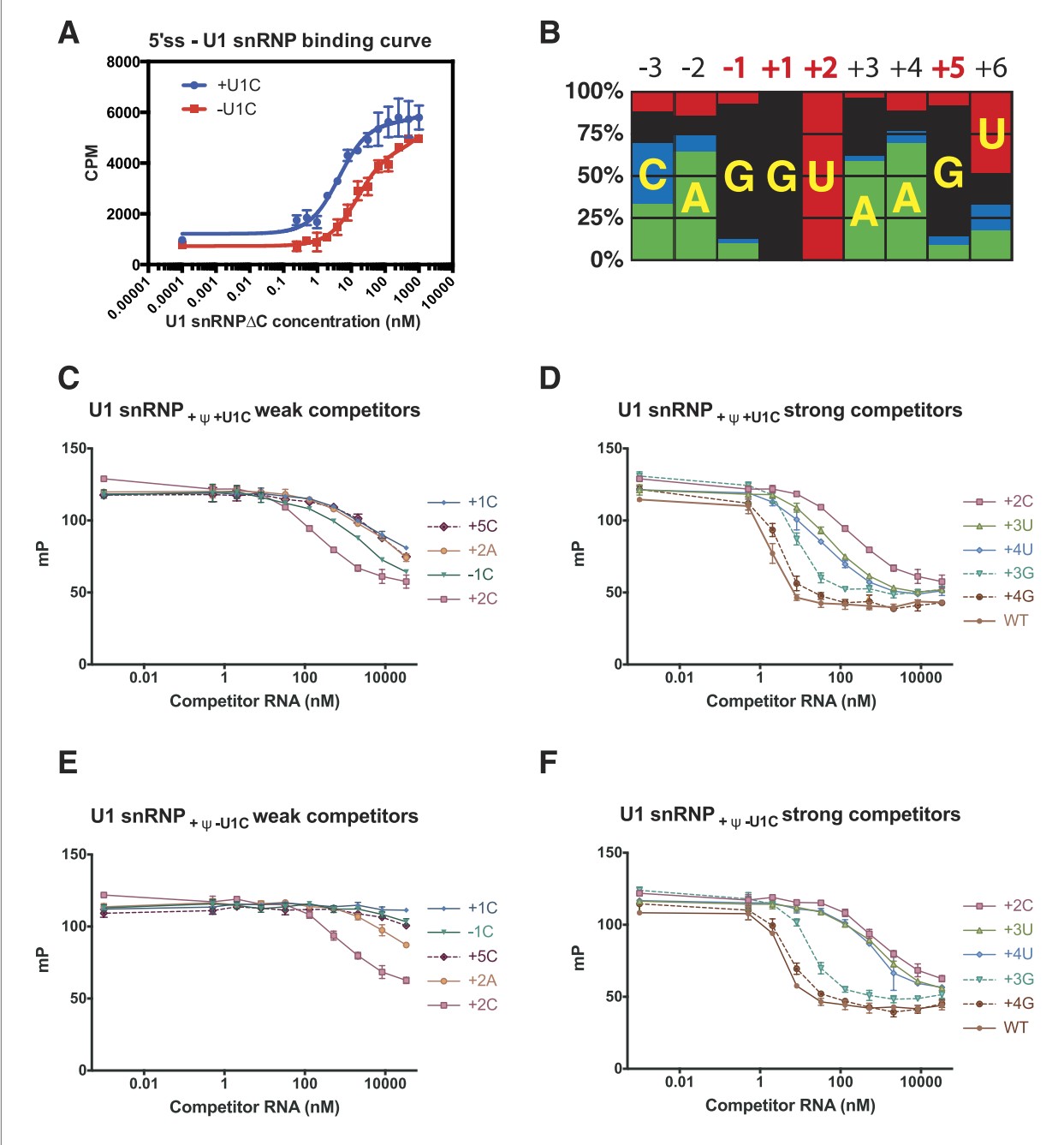

**Figure 7**. Influence of nucleotide substitutions at the 5′-splice site on U1 snRNP binding. (**A**) Filter-binding results for U1 snRNP reconstituted with and without U1-C to [$^{32}$P]-labelled 5′ splice site oligonucleotide. By curve fitting, the Kd with U1-C is 4.7 ± 0.8 nM and without U1-C is 15.8 ± 2.5 nM. CPM, counts per minute. (**B**) Nucleotides found at each position of the 5′-splice site of the U2-type introns. Adapted from *Roca et al. (2008)*. A, green; C, blue; G, black; U, red. Numbers for highly conserved positions are highlighted in red. (**C**) Competition assays of mutant 5′SS RNA binding to U1 snRNP containing U1C and uncapped but fully modified U1 snRNA. The 5′SS oligonucleotide with +1C, +5C, −1C and +2C substitutions compete weakly with the wild type oligonucleotide. In panels **C–F**, mP is an arbitrary unit of fluorescence polarization and error bars indicate standard error. (**D**) Competition assay with 5′SS oligonucleotides with +3G, +4G, +3U, +4U substitution and the wild type. 5′SS oligonucleotide with +2C substitution is included for comparison. (**E**) Same as in **B** except that U1 snRNP lacks U1-C. (**F**) Same as in **C** except that U1 snRNP lacks U1-C.

It has been proposed that U1 snRNP selects the same 5′SS sequence even in the absence of the 5′-end of U1 snRNA (*Du and Rosbash, 2002*; *Lund and Kjems, 2002*), and that U1C in isolation can recognize the 5′SS sequence. This conclusion is mainly based on a SELEX experiment, which may have

**Table 2.** 5′splice site binding

| Oligo name | Sequence* |
|---|---|
| **5ss-F** | **AGGAAAGUAU-F†** |
| WT | CAAAGGUAAGUUGGA |
| −1C | CAAA**C**GUAAGUUGGA |
| +1C | CAAAG**C**UAAGUUGGA |
| +2A | CAAAGG**A**AAGUUGGA |
| +2C | CAAAGG**C**AAGUUGGA |
| +3U | CAAAGGU**U**AGUUGGA |
| +3G | CAAAGGU**G**AGUUGGA |
| +4U | CAAAGGUA**U**GUUGGA |
| +4G | CAAAGGUA**G**GUUGGA |
| +5C | CAAAGGUAA**C**UUGGA |

*Bold nucleotides highlight the position of mismatch.
†F denotes 3′-fluorescein label.

been prone to an artefact caused by incomplete removal of the 5′-end of U1 snRNA. We observed no binding of 5′SS oligonucleotide by human U1-C protein, which was shown to be properly folded by NMR (*Muto et al., 2004*). *Schwer and Shuman (2014)* investigated the role of conserved basic and hydrophilic residues in yeast U1-C: they mutated residues that form hydrogen bonds with the backbone atoms of either 5′SS (Thr11, Thr14, Tyr12 and His24) or U1 snRNA (His15 and Ser19), form a salt-bridge with SmD3 (Arg21) or lie near the phosphate backbone (Lys22 and Arg28 [Lys28 in yeast]) in our crystal structure (*Figure 4*). Substitution of any one of these residues with Ala had no effect on yeast growth at any temperature indicating that these mutations do not have a major influence on 5′SS selectivity, in agreement with our conclusion.

## Conclusions

The structures of two sub-domains of U1 snRNP reported here have revealed an intricate network of interactions between the components of U1 snRNP. U1-70k N-terminal peptide binds to the subunit interfaces between SmD2 and SmF, and between SmD3 and SmB, and hence only the fully-formed core domain can induce its binding, which in turn enables U1-C to bind the core domain. Our structure also revealed the molecular contacts between U1 snRNP and the 5′SS of pre-mRNA and provided new insights into the molecular mechanism of 5′SS selection. U1-C makes no contacts with nucleotide bases and U1 snRNP selects 5′SS sequences primarily by thermodynamic stability of the RNA duplex between the 5′-end of U1 snRNA and the 5′SS. However U1-C fine-tunes the affinity to stabilize the binding of some mismatched 5′SS oligonucleotides relative to the canonical 5′SS.

## Materials and methods

### Protein preparation

The SmE/SmF/SmG trimer, the SmD1/SmD2 dimer, the SmB/SmD3 dimer, U1-A, U1-70k and U1-C were prepared as described previously (*Oubridge et al., 1994*; *Kambach et al., 1999*; *Muto et al., 2004*; *Pomeranz Krummel et al., 2009*; *Leung et al., 2011*). All proteins are based on human sequences, unless otherwise indicated. The coding sequence of thioredoxin, (His)$_6$-tag, tobacco etch virus (TEV) protease cleavage site, and a U1-70K fragment (residues Thr 2–Arg 59) was PCR-amplified from the U1-70K expression vector and inserted at the initiation codon of SmD1 in the SmD1/SmD2 coexpression vector (*Kambach et al., 1999*) to create the U1-70kSmD1/SmD2 expression vector (*Figure 1—figure supplement 1*). A (Gly–Ser)$_3$ sequence was included in the PCR primer to link the U1-70k fragment to SmD1. *Escherichia coli* BL21 (DE3) pLysS cells were transformed with the expression vector (*Studier et al., 1990*). The cells were grown at 37°C in 2xTY medium with ampicillin (100 µg/ml) and, when $A_{600 \text{ nm}} = 0.4 − 0.8$, protein expression was induced by addition of 0.5 mM IPTG and the culture was continued at 15°C overnight. Cell pellets were resuspended in Ni-A buffer (20 mM Tris-Cl pH7.4, 1 M NaCl, 1 M urea, 10 mM 2-mercaptoethanol) supplemented with EDTA-free protease inhibitor cocktail (Roche, Basel, Switzerland). The cells were lysed by sonication and clarified lysate was loaded onto a Ni-NTA column. The protein was eluted with a gradient of imidazole to 300 mM. The thioredoxin and (His)$_6$-tags of the protein were cleaved by His-tagged TEV protease during dialysis against Ni-A buffer before passing through the Ni-NTA column again. The flowthrough fractions were diluted fivefold with Na-0 buffer (20 mM Tris-Cl pH7.4, 1 M urea, 10 mM 2-mercaptoethanol) and loaded onto a HiTrap heparin column (GE Healthcare, Little Chalfont, UK) equilibrated with heparin-A buffer (20 mM Tris pH7.4, 200 mM NaCl, 1 M urea, 10 mM 2-mercaptoethanol). The U1-70kSmD1/SmD2 heterodimer was eluted by a NaCl gradient and peak fractions were pooled, concentrated to ~300 µM, rapidly frozen in liquid nitrogen and stored at −80°C.

A coding sequence for U1A70kF, (*Figure 1—figure supplement 1B*) consisting of a $(His)_6$-tag, TEV-protease cleavage site, residues 2–111 of U1-A protein, a linker of six Gly–Ser repeats and residues 60–216 of U1-70k protein, was constructed from PCR fragments and synthetic oligonucleotides, and ligated into the pET13 vector. *E. coli* BL21 (DE3) pLysS cells were transformed with the expression vector (*Studier et al., 1990*) and cells were grown in 2xTY medium with ampicillin (50 µg/ml) and chloramphenicol (34 µg/ml) at 37°C until $A_{600\ nm}$ was approximately 0.7. Then, the temperature was lowered to 20°C and protein expression induced by addition of 0.5 mM IPTG. After 8–10 hr cells were harvested by centrifugation and resuspended in lysis buffer (20 mM $Na^+$-Hepes pH 7.5, 25 mM imidazole, 0.5 M NaCl, 0.5 M urea). The cells were lysed by sonication, clarified and loaded onto a Ni-NTA column. The U1A70kF protein was eluted by a linear gradient of 25–500 mM imidazole in the same buffer. The His-tag was cleaved off with His-tagged TEV protease and the uncleaved protein was removed by passing the protein through a second Ni-NTA column. The protein was loaded onto a heparin-Sepharose column and eluted with a linear NaCl gradient (120–1000 mM) in 20 mM $Na^+$-Hepes pH 7.5, 25 mM Imidazole, 60 mM Na Phosphate pH 7.4, 1 M urea. Peak fractions were concentrated by ultrafiltration, buffer exchanged into 20 mM Na.Hepes, 25 mM imidazole, 0.3 M NaCl, pH 7.5, rapidly frozen in liquid nitrogen and stored at −80°C.

## RNA preparation

DNA templates for in vitro transcription (*Figure 1—figure supplement 2*) were assembled by ligating overlapping oligonucleotides, which included the T7 promoter, into pUC18 vector. All RNAs are based on human sequences, unless otherwise indicated. Genes for full-length and minimal U1 snRNAs lacking the first 10 nucleotides were also cloned into pUC18 together with T7 promoter. These truncated RNAs were transcribed in the presence of 2 mM GMP to facilitate ligation of modified 5′ end oligonucleotides. The 5′ fragment of U1 snRNA with post-transcriptional modifications (5′-AmUmACΨΨACCU-3′ or 5′-AmUmACUUACCU-3′ where Ψ = pseudo-uridine, Am, Um = 2′-O-methyl nucleotides) were purchased from Dharmacon (GE Healthcare, Little Chalfont, UK) and ligated to the truncated U1 snRNAs by splint-assisted ligation with T4 DNA ligase (*Ohkubo et al., 2013*). The plasmid template preparation, in vitro transcription and purification of RNA were carried out as described (*Price et al., 1995*).

## In vitro reconstitution and purification of full-length U1 snRNP

The full-length U1 snRNA at 8 µM was incubated in 50 mM KCl at 80°C for 2 min and annealed by snap cooling on ice. Three Sm protein sub-complexes were diluted to 100 µM each in reconstitution buffer (RB: 250 mM KCl, 20 mM $K^+$-HEPES pH 7.5, 5 mM DTT). RNA was mixed with Sm proteins so that the solution contained 4 µM of RNA, 6 µM of each of the Sm proteins, 10 mM of DTT, and 40 units/ml of RNasin (Promega, Fitchburg, Wisconsin, USA) in RB. The mixture was incubated at 30°C for 30 min, and then at 37°C for 15 min. U1-70k was diluted to 12 µM with RB. The Sm protein-RNA complex was mixed with an equal volume of the U1-70k solution and incubated on ice for 15 min. U1-A protein was added to 4 µM final concentration. The final reconstitution mix contained 2 µM of RNA, 3 µM each of the Sm proteins, 4 µM of U1-A, and 6 µM of U1-70k protein in RB. The solution was incubated on ice overnight and applied to a monoQ column (GE Healthcare, Little Chalfont, UK). The reconstituted complex was eluted with a gradient of KCl from 250 mM to 1 M in RB buffer. U1-C protein was added to U1 snRNP by buffers supplemented with 1.5 µM of U1-C protein to ensure an excess of U1-C in binding assays.

## In vitro reconstitution of minimal U1 snRNP

2 µM of SmKCm RNA (*Figure 1—figure supplement 2B*) was mixed with 1.5-fold molar excess of U1-70kSmD1/SmD2, SmD3/SmB and SmE/SmF/SmG sub-complexes in RB supplemented with 2 M urea. After incubation at 37°C for 1 hr, the sample was dialyzed against RB buffer at 4°C overnight. U1-C protein was added to a final concentration of 6 µM and incubated at 30°C for 15 min and at 4°C for at least 1 hr. The complex was purified on a monoQ column as described for full-length U1 snRNP.

## In vitro reconstitution of U1A70kF complex with SL1·SL2 RNA

U1A70kF protein was added slowly to 10 µM SL1·SL2 RNA in 0.4 M NaCl, 40 mM $Na^+$-Hepes pH 7.5, 50 mM imidazole at room temperature for a final protein concentration of 15 µM. After 20 min the mixture was diluted with an equal volume of water and incubated for a further 10 min. The complex was purified on a monoQ column equilibrated with 20 mM $Na^+$-Hepes pH 7.5, 150 mM NaCl and

eluted with a NaCl gradient (150 mM–1 M). Peak fractions were pooled, concentrated to 6 mg/ml with a centrifugal concentrator, and buffer exchanged into 0.2 M NaCl, 20 mM Na$^+$-Hepes pH 7.5, 25 mM imidazole.

## Crystallization of the minimal U1 domain

Crystals of the SmKCm complex were obtained by sitting-drop vapour diffusion at 277K. The purified U1 snRNP complex was mixed with 1.2-fold molar excess of 5'SS RNA oligo (5'-AGGUAAGUCC-3') purchased from Dharmacon (GE Healthcare, Little Chalfont, UK) and 0.15 mM of polyamine-9 (*Sauter et al., 1999*). The complex solution (4 mg/ml in 300 mM KCl, 20 mM K$^+$-Hepes, pH 7.5, 5 mM MgCl$_2$, 1 mM DTT) was mixed with an equal volume of reservoir solution (7% MPD, 180 mM KCl, 5 mM MgSO$_4$, 50 mM Na$^+$-Hepes pH 6.4). The crystals were improved by streak-seeding from another crystallization drop using a feline whisker 3 min after mixing the drops. Crystals suitable for data collection appeared after 1 month, reaching maximum dimensions of 0.35 × 0.05 × 0.05 mm$^3$. Crystals were transferred to cryo-protection buffer (20% MPD, 200 mM KCl, 5 mM MgSO$_4$, 50 mM Na$^+$-Hepes pH 6.4, 25% PEG4000) in three steps with 30 min equilibration after each transfer and flash frozen in liquid nitrogen.

## Crystallization of the U1A70kF RNA complex

The U1A70kF RNA complex was mixed with an equal volume of 40% MPD, 0.15 M NaCl, 0.1 M sodium acetate pH 4.2 and equilibrated with the same solution by sitting drop vapour diffusion at 20°C. Crystals of type I (wedge-shaped) grew within 16–48 hr whereas crystals of type II (plate-like) grew in the same drops after several weeks. Only the latter type diffracted to sufficient resolution for a structure determination.

## Structure determination

Diffraction data were collected at Diamond Light Source beamlines I02, I03, I04 and I04-1. Data were integrated in iMosflm (*Leslie and Powell, 2007*) and scaled with Scala (*Collaborative Computational Project Number 4, 1994*). For the minimal U1 snRNP (SmKCm complex) an initial molecular replacement solution was found in Phaser (*McCoy et al., 2007*) using the U4 snRNP core domain proteins (*Leung et al., 2011*). These form similar ring structures in both U1 and U4 snRNP, and have a total mass of ~70 kDa, which is more than half the mass of the minimal U1 snRNP (~120 kDa). Phases were improved by density modification using Parrot (*Cowtan, 2010*). This allowed us to build the remainder of the model using Coot (*Emsley et al., 2010*). The N-terminal region of U1-70k was built de novo and we manually placed the zinc finger domain of U1-C protein (*Muto et al., 2004*) and HIV1 kissing loop structure (*Ennifar et al., 2001*) into electron density. The structure was refined by Refmac (*Murshudov et al., 1997*) (*Table 1*). For the U1A70kF-RNA complex, an initial molecular replacement solution was found in Phaser (*McCoy et al., 2007*) using part of the structure of U1-A complex with SL2 RNA (*Oubridge et al., 1994*) and a homology model of the U1-70k RRM, which was built with Modeller (*Mart-Renom et al., 2000*) using multiple templates and manually refined alignment as an input. The remainder of the model was built into the MR map using Coot (*Emsley et al., 2010*), and the structure was refined by Refmac (*Murshudov et al., 1997*) (*Table 1*). Figures of molecular structures were drawn using Pymol (www.pymol.org).

## Pre-mRNA binding assays

Pre-mRNA oligonucleotide substrates containing 5' splice-site sequences were purchased from Dharmacon (GE Healthcare, Little Chalfont, UK) (*Table 2*). All the components were diluted in binding assay buffer (BAB: 10 mM K$^+$-Hepes pH7.5, 200 mM KCl, 2 mM MgCl$_2$, 0.5 mM DTT, 100 µg/ml tRNA, 50 µg/ml BSA), and 1.5 µM U1-C protein was added to BAB to ensure saturation of U1 snRNP with U1-C protein under assay conditions. In competition assays, each reaction (16 µl) containing 1 nM fluorescein-labeled reference RNA oligo (5ss-F) and 35 nM U1 snRNP reconstituted with U1 + Ψ RNA (*Figure 1—figure supplement 2*), was titrated with non-labelled competitor 5'SS RNA oligos. Binding curves were measured by fluorescence polarization using Pherastar (BMG Labtech, Ortenberg, Germany). The assays were carried out in triplicate. The values of fluorescence polarization in the absence of competitor RNA oligo were plotted at 1 pM for reference.

In the filter-binding assay, a 5'SS RNA oligonucleotide (5'-$^{32}$P- CAAAGGUAAGAUGGA-3', 10 fmol), was mixed with various concentrations of U1 snRNP, with or without U1-C protein, in Filter-Binding Buffer (FBB: 200 mM KCl, 2 mM MgCl$_2$, 0.5 mM DTT, 10 mM Hepes, pH 7.5, 100 µg/ml tRNA, 50 µg/ml BSA) in a final volume of 40 µl and incubated at 22°C for 2 hr. The binding reaction was passed, under

vacuum, through a Schleicher & Schuell (Dassel, Germany) NC45, 25 mm diameter filter that had been pre-wetted with FBB. The filter was washed with 0.5 ml FBB and then dried. The proportion of radiolabeled oligonucleotide retained on the filter was determined by scintillation counting. The experiment was performed in triplicate.

## Acknowledgements

We thank our former colleagues, particularly Daniel Pomeranz Krummel, Christian Kambach, Tijana Ignjatovic, and Yutaka Muto, who contributed to the early stages of this project, Chris Johnson for his invaluable advice and help with RNA binding assays. We also thank Jade Li, Adelaine Leung, Andy Newman, Wojtek Galej, Kelly Nguyen, Pei-Chun Lin, Garib Murshudov, Rob Nicholls, Fei Long, Paul Emsley, Tony Andreeva and Minmin Yu for help and support. We thank beamline staff at DIAMOND Light Source. This project was funded by the Medical Research Council. Yasushi Kondo was funded by the Nakajima Foundation during his PhD. Structure factors and atomic coordinates have been deposited with the PDB (entry codes 4PJO, 4PKD).

## Additional information

### Funding

| Funder | Grant reference number | Author |
|---|---|---|
| Medical Research Council | U105184330 | Kiyoshi Nagai |

The funder had no role in study design, data collection and interpretation, or the decision to submit the work for publication.

### Author contributions

YK, Attempted to crystallize the whole of U1 snRNP, Crystallized, Determined the structure of the minimal U1 snRNP complex, Performed the 5′SS binding assay and analysed the data, Contributed to the manuscript; CO, Attempted to crystallize the whole of U1 snRNP and refine the structure of the minimal U1 snRNP, Crystallized, Determined and refined the structure of the U1-70k-stem-loop I complex, Contributed to the manuscript; A-MMR, Attempted to crystallize the whole of U1 snRNP, Contributed to the manuscript; KN, Initiated and contributed to the U1 snRNP project since the beginning, Contributed to the manuscript

## Additional files

### Major dataset

The following datasets were generated:

| Author(s) | Year | Dataset title | Dataset ID and/or URL | Database, license, and accessibility information |
|---|---|---|---|---|
| Kondo Y, Oubridge C, van Roon AM, Nagai K | 2014 | Minimal U1 snRNP | http://www.pdb.org/pdb/search/structidSearch.do?structureId=4pjo | Publicly available at NCBI Protein Data Bank. |
| Kondo Y, Oubridge C, van Roon AM, Nagai K | 2014 | U1-70k in complex with U1 snRNA stem-loops 1 and U1-A RRM in complex with stem-loop 2 | http://www.pdb.org/pdb/search/structidSearch.do?structureId=4pkd | Publicly available at NCBI Protein Data Bank. |

**Reporting Standards:** Standard used to collect data: Validation reports are included in the data base.
The following previously published dataset was used:

| Author(s) | Year | Dataset title | Dataset ID and/or URL | Database, license, and accessibility information |
|---|---|---|---|---|
| Leung AKW, Nagai K, Li J | 2014 | Spliceosomal U4 snRNP core domain | http://www.rcsb.org/pdb/search/structidSearch.do?structureId=4WZJ | Publicly available at NCBI Protein Data Bank. |

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
