## [Decision Letter]

Thank you for sending your work entitled “Crystal structure of human U1 snRNP
reveals the mechanism of 5' splice site recognition” for consideration at
*eLife*. Your article has been favorably evaluated by James Manley
(Senior editor) and 3 reviewers, one of whom is a member of our Board of Reviewing
Editors.

The following individuals responsible for the peer review of your submission have agreed
to reveal their identity: Timothy Nilsen (Reviewing editor); Elena Conti and Joan Steitz
(peer reviewers).

The Reviewing editor and the other reviewers discussed their comments before we reached
this decision, and the Reviewing editor has assembled the following comments to help you
prepare a revised submission.

As you will see, two of the referees thought that the work was very nice and could be
published as is. The third referee agrees that the work is very important and is worthy
of publication in *eLife*. However they raise a number of points that
must be dealt with via revision. Please address each of these points as thoroughly as
possible. When revision is complete resubmit electronically together with a letter
detailing all changes.

Reviewer #1:

This manuscript reports high resolution x-ray structures of two sub-domains of U1 snRNP.
The two structures reveal unprecedented insight into the organization of the RNP as well
as the mechanism by which this RNP recognizes 5' splice sites. Both structures
contain a wealth of information regarding protein-protein interactions as well as
protein-RNA interactions present in the RNP. Perhaps the most surprising aspect of the
paper is that the U1-C protein does not make base specific contacts with the 5'
splice site. This finding dispels the currently accepted notion that U1-C participates
in 5' splice site recognition. Overall, the manuscript is exceptionally clearly
presented and the results are highly significant. Publication in *eLife*
is recommended as is.

*Reviewer #2*:

The manuscript by Kondo et al. elucidates the mechanism of 5' splice site
recognition. Based on their previous work, the authors cleverly engineered two
subcomplexes of the human U1 snRNP and determined their corresponding structures at high
resolution. The fundamental importance of this work is that the results explain how the
5' splice site of a pre-mRNA is recognized by the U1 snRNP. The work is thorough,
the data are of high quality, the figures and text are clear, and the message is very
important. This is a beautiful piece of work.

Reviewer #3:

Kondo et al used X-ray crystallography to solve two structures related to the human U1
snRNP. Their structural data conclusively show that the 5′ splice site is
recognized via base pairing with U1 snRNA and not another protein in the snRNP complex.
Furthermore, the structures establish a network of interactions within the U1 snRNP,
particularly U1 snRNA, U1-C, U1-70k and the seven Sm proteins. The authors also employ
binding assays to validate the importance of base pairing between U1 snRNA and RNA
containing the 5′ splice site.

Overall, the work rests on a long-time controversy in the splicing field; whether the
sequence of the 5'-splice site is recognized by both an RNA and a protein and is
therefore worthy of publication in *eLife*. However, several scientific
issues need to be addressed and the clarity of the presentation requires significant
improvement.

Scientific comments:

1) Pre-mRNA binding assays: For oligo 5ss-F, the sequence does not appear to be a
consensus 5'-splice site. Thus, the competing oligos have a consensus splice site
but the sequence differs by more than a single-nucleotide position. The authors need to
explain why these sequences were used or repeat the assay using oligos with the proper
sequence.

2) The main text provides no details about how the crystal structures were determined,
i.e. by molecular replacement. The rationale for using the U4 snRNP core domain as a
model was not explained. Is it simply because the Sm proteins are shared?

General comments:

Interactions are described extensively throughout this manuscript. However, specific
interactions (i.e. amino acid residues and nucleotides) are usually not stated. If
specific interactions are mentioned, they often are not labeled in the figures, making
it difficult to understand the authors' interpretation of the structure. More
details need to be added to the figure legends to accurately and precisely describe the
interactions shown in the figures.

Minor comments:

Abstract:

The resolution of the crystal structures (3.3 and 2.5 Ang) needs to be added. Also,
“subdomains” is misleading because this term is not generally applied to
the U1 snRNP. Perhaps the authors should use “synthetic substructures”
instead.

Introduction:

The range of nucleotides in the invariant sequence is not mentioned. The authors should
insert “(nts #-#)” to specify the nucleotides of interest?

The Introduction compares 5′SS recognition in yeast and humans. It appears the
authors are trying to make the point that this work is focused on the conserved regions.
To make this point more clear, perhaps the authors should begin by explaining how the
yeast and human U1 snRNPs differ. Although “human” is used in the Title,
the authors should reconfirm that all the components in this work are based on human nt
sequences and proteins, correct?

The Introduction does not end with a short summary of the major findings of this work,
leaving the reader unsatisfied. The last paragraph of the Introduction is a run-on
paragraph. The authors should divide the paragraph into two independent paragraphs: one
for the yeast-human comparison and one for the reported findings.

The Introduction talks about the role of U1 in splicing. However, there is no mention of
its other important cellular function, telescripting.

Results:

The text mentions seven Sm proteins but Figure 1
labels only five Sm proteins.

“The overall structure of the core domain is very similar to that of the U4 snRNP
core domain.” Please include rmsd to support the descriptor “very
similar”. Also, how do you know the similarity is not due to model bias?

Helix H0 and helix H are two examples of features being mentioned that are not
shown/labeled.

What are the “key residues”? The residues are also not identified in the
figure legend. The reader must guess what residue is at the “equivalent
position”. Where are loops L3 and L5?

The authors talk about nucleotides yet the exception they list, SmD1, is a protein. This
is confusing.

Binding is stabilized by salt bridges and hydrogen bonds – what residues and/or
nucleotides participate?

Because U4 is referred to several times throughout this manuscript, a picture of U4
should be shown in the supplement, explaining what components constitute the
“core”.

The text specifically identifies strands and helices of importance but these secondary
structures are not labeled in the figures (e.g. beta 4 strand in Figures 3 and 4). There are other such omissions throughout
the manuscript.

Be quantitative about how much the R21Q mutation weakens binding.

Are pseudo-uridines present in both human and yeast snRNAs?

The “E complex” needs to be defined and its relevance explained for the
general reader.

“Conserved well” *–* what exactly is the percentage
of sequence similarity?

What is meant by the 11-nt loop effectively being a 5-nt loop? How does base stacking
reduce loop length?

C33 and G34 are not labeled in Figure 5. In Figure 5, the side chains of R191 and K138 should be
shown?

The filter binding assay is not adequately described in Materials and methods.

Materials and methods:

1) Are all proteins and nt sequences based on human?

2) Prior to the induction step, were cells grown at 15^°^C?

3) What is the final storage buffer? Was glycerol not added?

4) No storage conditions are described. Was the protein never frozen?

5) Steps are described for annealing full-length U1 snRNA but it is not stated to what
it is being annealed. Do the authors mean folding rather than annealing? Also, is the
annealing step performed in the absence of buffer?

Figures & legends:

The descriptions in the figure legends are inadequate. Many of the figures suffer from
the same problems. Please explain what is represented by colored objects and dashed
lines in each figure legend. The colors in Figure 1—figure supplement 1 are not consistent with Figure 1 (e.g. SmF is brown in Figure 1 but green in the supplement). Please add more information about
interactions between specific residues and nucleotides. Make sure key secondary
structures, nucleotides and amino acids mentioned in the text are labeled in the
figures, especially Figures 3 and 4.

Figure 1: For readers unfamiliar with the U1
snRNP, it would be helpful if there were a cartoon schematic for each crystal structure.
Then, it would be easier to visualize the location of each component. Also, Figure 1 could show representative examples of the
electron density for each structure. Then, Figure 2 could be deleted and replaced with Figure 2—figure supplement 1, which contains more substantial structural
information.

Figure 1—figure supplement 1: What are
the black, blue and red lines?

Figure 2: Is this a stereoview?

Figure 2—figure supplement 1: The legend
mentions electron density for A125 in panel A but no density is shown. Please add more
details about panels B-H; the amino acid residues are important as well as the
nucleotides.

Figure 4: What do the teal/pale pink colors
represent?

Figure 7: A description of the plots is missing.
What are “CPM” and “mP”? Do the error bars represent SD or
SE? What are the Kd values extrapolated from the data in Figure 7? The text mentions a 4-fold difference but doesn't state a
numerical value with units. In panels C and E, WT data are missing.

Tables:

In Table 2, is the sequence for -1C correct? It
seems that the red U should be a C.

---

## [Author Response]

*Scientific comments*:

*1) Pre-mRNA binding assays: For oligo 5ss-F, the sequence does not appear to be
a consensus 5'-splice site. Thus, the competing oligos have a consensus splice
site but the sequence differs by more than a single-nucleotide position. The authors
need to explain why these sequences were used or repeat the assay using oligos with
the proper sequence*.

We found that a labelled consensus sequence oligonucleotide bound too tightly to be
competed off by the weaker competitor oligonucleotides at concentrations that could be
feasibly achieved in the experiment. Hence we used a labelled oligonucleotide with one
mismatch (5SS-F). This does not influence the experiment's capacity to show whether
one oligonucleotide competes better or worse than another.

*2) The main text provides no details about how the crystal structures were
determined, i.e. by molecular replacement*.

We now make it clear in the main text that both structures were determined by molecular
replacement with a sentence added to the end of the first paragraph of the Results
section.

The rationale for using the U4 snRNP core domain as a model was not explained.
Is it simply because the Sm proteins are shared?

It is now explained in Materials and methods section.

General comments:

*Interactions are described extensively throughout this manuscript. However,
specific interactions (i.e. amino acid residues and nucleotides) are usually not
stated. If specific interactions are mentioned, they often are not labeled in the
figures, making it difficult to understand the authors' interpretation of the
structure. More details need to be added to the figure legends to accurately and
precisely describe the interactions shown in the figures*.

We have now labelled amino acid residues and nucleotide as requested in the figures.
Chains are coloured consistently throughout the manuscript.

Abstract:

*The resolution of the crystal structures (3.3 and 2.5 Ang) needs to be added.
Also, “subdomains” is misleading because this term is not generally
applied to the U1 snRNP. Perhaps the authors should use “synthetic
substructures” instead*.

The second sentence of the Abstract has been changed to “We present two crystal
structures, at 2.5 Å and 3.3 Å resolution, of engineered U1 sub-structures
which together reveal...” Although the term 'synthetic' was suggested
to describe the sub-structures, we believe the term “engineered” already
suffice to describe “something which was man-made”. Also, we now use the
hyphenated term 'sub-structure', rather than 'substructure'
throughout the text, because the latter only has the dictionary definition, 'an
underlying or supporting structure', which is not the intended meaning.

Introduction:

The range of nucleotides in the invariant sequence is not mentioned. The authors
should insert “(nts #-#)” to specify the nucleotides of
interest?

We inserted (nts1-10).

*The Introduction compares 5′SS recognition in yeast and humans. It
appears the authors are trying to make the point that this work is focused on the
conserved regions. To make this point more clear, perhaps the authors should begin by
explaining how the yeast and human U1 snRNPs differ. Although “human”
is used in the Title, the authors should reconfirm that all the components in this
work are based on human nt sequences and proteins*,
*correct?*

p5: To address several points made about the Introduction, an extra sentence has been
added to the 4th paragraph, comparing yeast and human U1 snRNP, and the nucleotide range
has been inserted in the subsequent sentence.

“Yeast U1 snRNP, when compared to the human particle, contains a larger and more
complex snRNA, which is associated with many protein factors (Prp39, Snu71, Prp40,
Prp42, Nam8, Snu56, Urn1 and Prp5), which have no counterparts in human U1 snRNP (66). However, despite
these differences, the sequence of the 5'-single stranded region of U1 snRNA (nts
1-10) is invariant from yeast to human”.

*The Introduction does not end with a short summary of the major findings of this
work, leaving the reader unsatisfied. The last paragraph of the Introduction is a
run-on paragraph. The authors should divide the paragraph into two independent
paragraphs: one for the yeast-human comparison and one for the reported
findings*.

A short summary paragraph has been added to the end of the Introduction.

“The two crystals together reveal the structures of the substantial parts of U1
snRNP at high resolution and provide crucial insights into the mechanism of pre-mRNA
recognition by U1 snRNP. In particular, we find that U1-C makes no base-specific
contacts with the 5'SS sequence. Also, by measuring the intrinsic affinity of
recombinant U1 snRNP for various 5’SS sequences we disentangle the role played by
the U1 snRNP from the other complexities of 5’SS recognition, and assess the
relative contributions of U1 snRNA and U1-C protein in light of our crystal
structure.”

*The Introduction talks about the role of U1 in splicing. However, there is no
mention of its other important cellular function, telescripting*.

An extra sentence has been added to the end of paragraph 1, mentioning the role of U1
snRNP in regulating cleavage and polyadenylation of mRNA (telescripting) with two
references.

“U1 snRNP is also an important regulator of mRNA 3' end cleavage and
polyadenylation ([1]; reviewed
in [59]).”

Results:

*The text mentions seven Sm proteins but*
Figure 1
*labels only five Sm proteins*.

We have now labelled all components in Figure 1.

“The overall structure of the core domain is very similar to that of the
U4 snRNP core domain.” Please include rmsd to support the descriptor
“very similar”. Also, how do you know the similarity is not due to
model bias?

We included rmsd in the text. The U4 snRNP structure has now been refined to Rfree of
22.4% and the U1 snRNP structure has been refined to Rfree of

25.5%. Hence model bias would not be an issue for ordered regions we discuss in this
manuscript. We included rmsd of the two structures in the text.

*Helix H0 and helix H are two examples of features being mentioned that are not
shown/labeled*.

It is shown in Figure 1 and we now refer to this
figure in the text.

*What are the “key residues”? The residues are also not identified
in the figure legend. The reader must guess what residue is at the “equivalent
position”*. *Where are loops L3 and L5?*

The key residues in loops L3 and L5 are now identified in Figure 2—figure supplement 1 legend.

*The authors talk about nucleotides yet the exception they list, SmD1, is a
protein. This is confusing*.

This has been changed to clarify that nucleotide G130 interacts differently with protein
SmD1.

*Binding is stabilized by salt bridges and hydrogen bonds* –
*what residues and/or nucleotides participate?*

We added “its binding to SmD2 is stabilized…” to clarify this
statement.

*Because U4 is referred to several times throughout this manuscript, a picture of
U4 should be shown in the supplement, explaining what components constitute the
“core”*.

We added “which consists of seven Sm proteins and U4 snRNA when the U4 core snRNP
first appears”.

*The text specifically identifies strands and helices of importance but these
secondary structures are not labeled in the figures (e.g. beta 4 strand in*
Figures 3 and 4*).
There are other such omissions throughout the manuscript*.

See Figure 3. We do not consider it necessary to
label the 3_10_ helix of U1-70k in Figure 4 because this helix is defined as residues 6-12 in the text and the start
and end residues (Pro6 and Leu12) are labeled in the figure. Furthermore, this region of
the figure is already quite crowded.

*Be quantitative about how much the R21Q mutation weakens binding*.

We added “by about 10-fold” to this sentence.

*Are pseudo-uridines present in both human and yeast snRNAs*?

Pseudo-uridines are conserved at the 5' end of U1 snRNA of yeast and human. The
following has been added to paragraph 4 of the Results section and a supporting
reference has been added to References (see below).

“Pseudo-uridines at these positions are conserved from yeast to human U1 snRNA
(31).”

*The “E complex” needs to be defined and its relevance explained
for the general reader*.

“At an early stage of spliceosomal assembly U1 snRNP binds to the 5’SS and
U2 auxiliary factor (U2AF) and splicing factor 1 (SF1) bind to the 3’ splice site
and branch point resulting in the formation of E complex.” This has been inserted
before the sentence to define the E complex.

“Conserved well” – what exactly is the percentage of
sequence similarity?

“The first 215 residues of U1-70k are conserved well from yeast to human (48%
sequence similarity), suggesting that this region has an evolutionarily conserved
essential function”.

What is meant by the 11-nt loop effectively being a 5-nt loop? How does base
stacking reduce loop length?

This is illustrated in Figure 5.

*C33 and G34 are not labeled in*
Figure 5. *In*
Figure 5*, the side chains
of R191 and K138 should be shown?*

Labels for C33 and G34 and the side chains of R191 and K138 have been added to Figure 5.

*The filter binding assay is not adequately described in Materials and
methods*.

See Materials and methods, end of final section.

'Mechanism of 5'SS recognition', first paragraph: to be more accurate
about U1 snRNP affinity for 5'SS RNA ±U1-C, the penultimate sentence has been
altered.

“The affinity of U1 snRNP without U1-C (U1 snRNP( U1-C)) for the wild type
5’SS oligonucleotide increases by between three and four-fold on addition of U1-C
(Figure 7).”

Materials and methods:

1) Are all proteins and nt sequences based on human?

“All proteins are based on human sequences, unless otherwise indicated”
was added as the second sentence in the Protein Preparation section and the sentence.
“All RNAs are based on human sequences, unless otherwise indicated” was
added to the RNA Preparation section.

*2) Prior to the induction step, were cells grown at
15*^*°*^*C?*

The growth temperature prior to induction has been added.

“The cells were grown at 37°C in 2xTY medium with ampicillin (100
∝g/ml) and, when A_600 nm_ = 0.4-0.8, protein expression was
induced by addition of 0.5 mM IPTG and the culture was continued at 15°C
overnight.”

3) What is the final storage buffer? Was glycerol not added?

The given method is correct; no glycerol was added. However, the complex was
flash-frozen in liquid nitrogen before storage and an explanatory clause has been
added.

“The U1-70kSmD1/SmD2 heterodimer was eluted by a NaCl gradient and peak fractions
µM, were pooled, concentrated to ∼300 rapidly frozen in liquid nitrogen and
stored at -80°C.”

4) No storage conditions are described. Was the protein never
frozen?

Storage conditions have been added as follows:

“Peak fractions were concentrated by ultrafiltration, buffer exchanged into 20 mM
Na.Hepes, 25 mM imidazole, 0.3 M NaCl, pH 7.5, rapidly frozen in liquid nitrogen and
stored at -80°C.”

5) Steps are described for annealing full-length U1 snRNA but it is not stated
to what it is being annealed. Do the authors mean folding rather than annealing?
Also, is the annealing step performed in the absence of buffer?

The term annealing is commonly used to refer to the process of folding RNA by heating
and subsequent cooling. We do not believe there is any ambiguity here. It is correct
that the annealing was performed in the absence of buffer; optimal conditions were found
empirically to be in 50 mM KCl.

End of final section: Added description of Filter-Binding Assay.

“In the filter-binding assay, a 5'SS RNA oligonucleotide
(5'-^32^P-CAAAGGUAAGAUGGA-3', 10 fmol), was mixed with various
concentrations of U1 snRNP, with or without U1-C protein, in Filter Binding Buffer (FBB:
200mM KCl, 2 mM MgCl_2_, 0.5 mM DTT,µg/mL10tRNA,mM Hepes, pH 7.5, 100
50µg/mL BSA) in a final volume of 40 µl and incubated at 22°C for 2
hours.

The binding reaction was passed, under vacuum, through a Schleicher & Schuell NC45,
25 mm diameter filter that had been pre-wetted with FBB. The filter was washed with 0.5
mL FBB and then dried. The proportion of radiolabeled oligonucleotide retained on the
filter was determined by scintillation counting. The experiment was performed in
triplicate.”

Figures & legends:

*The descriptions in the figure legends are inadequate. Many of the figures
suffer from the same problems. Please explain what is represented by colored objects
and dashed lines in each figure legend. The colors in*
Figure 1—figure supplement 1
*are not consistent with*
Figure 1
*(e.g. SmF is brown in*
Figure 1
*but green in the supplement). Please add more information about interactions
between specific residues and nucleotides. Make sure key secondary structures,
nucleotides and amino acids mentioned in the text are labeled in the figures,
especially*
Figures 3 and 4.

Figure 1*: For readers
unfamiliar with the U1 snRNP, it would be helpful if there were a cartoon schematic
for each crystal structure. Then, it would be easier to visualize the location of
each component. Also,*
Figure 1
*could show representative examples of the electron density for each structure.
Then,*
Figure 2
*could be deleted and replaced with*
Figure 2—figure supplement 1*, which contains more substantial structural
information*.

Figure 1: Labels for SmB and SmD1 were not put
on this figure because these proteins are mostly concealed at the back of the Sm ring.
Labels have now been added to indicate their position and color-coding. The positions of
RNA Helix H and SmD2 helix H0 are now also indicated on the figure.

Figure 1 legend: This sentence has been added to
part A to explain the labels H and H0 in panel A.

“Label H indicates U1 snRNA helix H; H_0_ indicates the first alpha
helix of SmD2 protein.”

Figure 1—figure supplement 1*: What are the black, blue and red lines?*

In response to the general comment, this figure has been re-colored to be properly
consistent with Figure 1.

Figure 1—figure supplement 1 legend:

“The thinner lines below the bars represent the constructs used for
crystallization: blue lines for sequences used in the U1A70kF fusion protein, red lines
for sequences used in the 70kSmD1F fusion protein, and black lines indicate the extent
of protein constructs used in minimal U1 crystallisation.”

Figure 2*: Is this a
stereoview?*

First line now indicates that this is a stereoview.

“Stereoview showing binding of U1 snRNA at the central hole of the Sm protein
assembly.”

Figure 2—figure supplement 1*: The legend mentions electron density for A125 in panel A
but no density is shown. Please add more details about panels B-H; the amino acid
residues are important as well as the nucleotides*.

This sentence has been moved into the Figure 2
legend from the legend for Figure 2—figure supplement 1 and altered slightly for clarity.

“Electron density for A125 and the phenol ring of SmF Y39, which stacks on it, is
weak.”

Figure 2—figure supplement 1 legend: Has
been re-written to address the comments about loops L3 and L5 and 'key
residues' made regarding Results section.

Figure 4*: What do the
teal/pale pink colors represent?*

The meaning of the teal and pale pink (fawn) nucleotides is explained.

“(B) U1-C forms hydrogen bonds with the sugar-phosphate backbone atoms but makes
no contact with RNA bases. On the 5'SS strand, nucleotides are colored teal for
exonic and fawn for intronic sequence. (C) Schematic representation of the
5'-splice site recognition. Red dotted lines, hydrogen bonds made by amino acid
side chains of U1-C; blue dotted lines, hydrogen bonds made by main chain atoms of U1-C.
The 5'SS nucleotides are color-coded as in panel B.”

Figure 7*: A description
of the plots is missing. What are “CPM” and “mP”? Do the
error bars represent SD or SE? What are the Kd values extrapolated from the data
in*
Figure 7*? The text
mentions a 4-fold difference but doesn't state a numerical value with units. In
panels C and E, WT data are missing*.

Figure 7 legend: Made clear that the graph is of
the results of a filter-binding experiment, added Kd values from filter-binding and
defined 'CPM'.

“(A) Filter-binding results for U1 snRNP reconstituted with and without U1-C to
[^32^P]-labelled 5’splice site oligonucleotide. By curve fitting, the
Kd with U1-C is 4.7 ± 0.8 nM and without U1-C is 15.8 ± 2.5 nM. CPM, counts
per minute.”

Figure 7 legend: A sentence has been added to
the end of this section of the legend defining 'mP' and explaining the error
bars.

“In panels C-F, mP is an arbitrary unit of fluorescence polarization and error
bars indicate standard error.”

Instead of WT, we use the +2C data to compare Figure 7 and Figure 7;
this prevents the figures from looking too crowded.

Tables:

*In*
Table 2*, is the
sequence for -1C correct? It seems that the red U should be a C*.

The sequence for -1C was indeed incorrect. This row has been amended. We have also
removed rows -2U and +6A because these sequences weren't used in the
experiments reported in this paper. A wild-type (WT) row has been added for the
reader's reference.